# Precise Crop Pest Detection Based on Co-Ordinate-Attention-Based Feature Pyramid Module

**DOI:** 10.3390/insects16010103

**Published:** 2025-01-20

**Authors:** Chenrui Kang, Lin Jiao, Kang Liu, Zhigui Liu, Rujing Wang

**Affiliations:** 1School of Information Engineering, Southwest University of Science and Technology, Mianyang 621010, China; 18108199160@139.com (C.K.); liuzhigui@swust.edu.cn (Z.L.); 2Institute of Intelligent Machines, Hefei Institutes of Physical Science, Chinese Academy of Sciences, Hefei 230031, China; 3School of InterNet, the National Engineering Research Center for Agro-Ecological Big Data Analysis & Application, Anhui University, Hefei 230031, China; rjwang@iim.ac.cn; 4Department of Aeronautical and Aviation Engineering, Hong Kong Polytechnic University, Hong Kong 999077, China; k1liu@polyu.edu.hk

**Keywords:** crop pest, object detection, small pest, feature pyramid network, co-ordinate attention, sample selection

## Abstract

Accurate detection of crop pests plays a vital role in integrated pest management (IPM). Traditionally, this task has been carried out through manual observation and counting in field environments, which is not only time-consuming but also requires a significant amount of labor. To overcome these limitations, by leveraging a deep convolutional neural network, a new crop pest detection module has been developed for automatic and accurate recognition and localization of crop pests. The proposed method has been tested on a large dataset of corn pests, and the results demonstrate its excellent performance in precisely recognizing and detecting crop pests.

## 1. Introduction

The negative effects of various agricultural pests result in a large reduction in crop production. In order to improve the quality and yield of grain, integrated pest management methods have been widely applied in agricultural production to reduce the impact of pests. The precise detection and counting of crop pests is a key step in integrated insect pest management, which can help growers to spray the appropriate amount of pesticides onto fields. The traditional methods adopt manual observation to obtain information about insect pests. However, this process is expensive (labor), time-consuming, has low efficiency, and has subjective counting results. Significant strides have been made in the domain of computer vision techniques for object recognition and detection, attaining remarkable success. Therefore, these detection methods have been introduced into crop pest detection and have shown significant advances. A computer-vision-based agricultural pest recognition method was proposed by merging a local model and global features [1]. Experiments were carried out on five types of moths using a 464-image dataset, demonstrating that the proposed method obtains an accuracy of 86.6%. The authors of [2] developed an automatic crop pest recognition method using unsupervised learning with multi-level feature maps. The classification results verified the advantages of the proposed methods. Although machine-learning-based pest detection methods have performed well in pest recognition and detection, the hand-crafted features need to be carefully designed according to different datasets to obtain precise detection results; this will result in impractical application in crop production scenarios. Deep-learning-based methods use convolutional neural networks (CNNs) for automatic feature extraction, which can alleviate the disadvantages of manual design features in machine learning methods. CNNs have been widely used for object detection and have achieved great success [3,4,5,6,7,8,9,10,11]. Currently, CNN-based object detectors contain single-stage [9,10,12,13] and multi-stage methods [3,14,15]. The former treat object detection as a single-shot issue, obtaining the bounding boxes of objects directly, whereas the latter require region proposals to ensure the precision of object detection. Because of the powerful feature extraction processes of CNNs, they have been introduced into the task of crop pest recognition. For instance, deep hybrid global- and local-activated features for automatic pest recognition using a localization module have been developed, and the experiments conducted on a large-scale pest dataset obtained an mAP of 75.03%, surpassing other detection approaches [16]. A deep-CNN-based module for recognizing pest images with 20 categories has been proposed [17], holding practical significance for intelligent pest identification in agriculture. The Visual Geometry Group Network (VGG16-Net) and InceptionV3 were applied to accomplish feature extraction, resulting in the accurate identification of rice diseases and pests. However, the above methods cannot accurately detect small-sized agricultural pests. Recently, multi-scale feature pyramid networks (FPNs) [5] that spread the semantic features from high levels into lower levels have become the mainstream method in small-object detection. They can be directly and effectively combined with any detector to achieve precise detection at various scales. Based on the structure of classical feature pyramid networks, many researchers have proposed various improved feature pyramid networks. For instance, a path aggregation network has been built by employing a bottom-up pathway to strengthen the localization signals in the lower layer [18]. Recently, by using multiple U-shaped networks, Zhou et al. constructed stronger feature pyramid representations [19]. However, the architectures of recent modules are hand-crafted. A method for neural architecture searches was employed to develop an innovative multi-scale feature pyramid network [20]. These methods have been used in pest detection. In order to extract the distinguishable features of small-sized pests, Dong et al. proposed a channel recalibration feature pyramid network and an adaptive anchor module to generate proper anchor boxes, leading to excellent performance in small-pest detection [21]. In order to alleviate the challenges in small-pest detection, Jiao et al. unitized an adaptive fusion factor instead of using a fixed one, leading to excellent performance in the detection of small pests. Simultaneously, an attention mechanism was introduced to address the problem of feature extraction for small objects. Kang et al. developed an attention-based feature fusion network for corn pest detection under wild environments, achieving 70.1% mAP and 74.3% recall. Multi-head self-attention has been used to capture feature dependencies over long distances regarding strawberry disease images by leveraging the self-attention mechanism. The experimental results on 12 types of strawberry disease datasets demonstrate that the recognition accuracy of the proposed method outperforms other methods. Additionally, numerous researchers contributed to improving the accuracy of small-pest detection regarding various aspects. An anchor-free region proposal generation network (AF-RPN) was designed for pest regions of interest [22], which obtained 56.4% for mAP and 85.1% for mRecall on a 24-class pest dataset, outperforming the existing methods at the time. In addition, to deal with the accurate detection of tiny and crowded pests, Arantza et al. introduced a deep-learning-based density estimation method for counting regression [23]. At the same time, Du et al. adopted a densely clustered tiny module to detect aphids of very small size and in high-density distributions. The other agricultural pest detection approaches, such as in [24,25,26,27], are almost all based on multi-stage object detectors or their improved versions. Nevertheless, the detection efficiency of multi-stage methods is low, which is impractical and infeasible in the agricultural field. Therefore, we need to explore faster and more precise agricultural pest detection methods. In order to improve detection efficiency and accuracy in small-pest detection, in this paper, we first introduce a co-ordinate attention block into the feature pyramid network to produce direction-aware and position-sensitive fused information. This results in the ability to augment the feature representations of instances of small pests. Additionally, for training the network, a soft loss function has been designed by using the generated negative and positive weights to optimize our CNN-based network. Finally, by using a state-of-the-art module, the fully convolutional one-stage detector, FCOS, as the baseline, extensive experimentation demonstrated that the proposed CAFPN and soft-weight loss function significantly enhanced the baseline FCOS performance, all without sacrificing inference speed. The main contributions of this study are described as follows:In order to achieve feature extraction of small-sized agricultural pests, a novel co-ordinate-attention-based feature pyramid network was designed, which can obtain direction-aware and position-sensitive features, leading to detection accuracy improvement.For the training of the pest detection module, a dynamic positive and negative pest sample selection strategy was devised by adopting a category-specific Gaussian-shaped function.Numerous comparative experiments using our constructed agricultural multi-category pest dataset indicate that our detection module demonstrates significant performance without a corresponding increase in computational burden.

## 2. Proposed Methods

### 2.1. Dataset

**AgriPest21 dataset:** In order to verify the performance of the proposed module, a large-scale multi-class agricultural pest dataset, AgriPest21, published in [26], was adopted. Figure 1 demonstrates the images of each type of cop pests in AgriPest21. As shown in Table 1, this dataset contains 21 types of crop pests with 24,412 images, with 22,000 images for optimizing the network and 2412 images for evaluating the availability of the detection modules. In order to improve detection speed, we resized the collected images to 800×600 pixels in this study.

We further analyzed the scale distribution of the AgriPest21 dataset. Figure 2 shows the distribution of the relative scale of all pest instances. Here, the relative scale is the proportion of the number of pest instances in the whole image. From Figure 2a, we can see that the distribution of the relative scales tends to be small. Figure 2b provides an example that illustrates the relative scale of the AsM pests, with only 0.015% occurrence. This poses large challenges for precise classification and localization. Furthermore, Figure 2c illustrates the density distribution of the AgriPest21 dataset, showing that all images have more than one pest instance, and about 20% of the images have more than five pest instances. Even worse, some images exhibit over forty pest instances, which causes poor performance regarding the pest detection module.

**IP102 dataset:** The pest images of IP102 are collected via the Internet [28]. There are 102 types of pests, and 75,222 images for image classification and 18,983 images for object detection task. In our work, we only selected 18,983 images for pest detection. Some images of IP102 dataset are shown in Figure 3. In addition, we have demonstrated related analysis regarding the IP102 dataset, as shown in Figure 4. From Figure 4a, we can see that there is a serious problem of imbalanced sample categories, there are various scales in IP102 dataset, and the relative scales of the pest targets tend to be large compared with our built AgriPest 21 dataset, as shown in Figure 4b. Some experiments are conducted on the IP102 dataset to evaluate the performance of method for detecting pests with various scales or under different backgrounds.

### 2.2. Proposed Method

This section provides a comprehensive overview of the intricate implementation details pertaining to our proposed detector, depicted in Figure 5a. Initially, we undertook a thorough review of the intricate network architecture of the feature pyramid network, examining its components and design principles. Subsequently, the co-ordination attention mechanism was introduced into feature fusion within the FPN network. Finally, we proposed a novel soft-weighted loss function aimed at mitigating false-positive predictions during training.

#### 2.2.1. Revisiting Feature Fusion in FPN

Two crucial factors, specifically the up-sampling ratio and the fusion ratio between adjacent layers, significantly impact the effectiveness of the feature pyramid network. While recent studies have successfully enhanced performance by decreasing the down-sampling ratio, this approach has inadvertently led to an increase in computational burden. To be specific, the operation of the feature pyramid network is represented as(1)$$Y_{i}=C_{i}\left(X_{i}\right)+F_{u}\left(Y_{i+1}\right)$$
where *C* represents the 1×1 convolutional operation for changing the number of channels. $F_{u}$ indicates the up-sampling process for enhancing resolution.

#### 2.2.2. Co-Ordinate-Attention-Based Feature Pyramid Network (CAFPN)

As we know, the feature fusion mechanism within an FPN simply involves adding the up-sampled higher-level feature maps. This approach ignores the positional information inherent in the feature maps, which holds significance in producing spatial attention weight maps. Therefore, a novel attention mechanism related to position information was developed, which can improve the module’s feature extraction ability. Figure 5c demonstrates the network structure of the CAFPN. Firstly, all feature maps generated by the deep residual network pass through the convolutional layer with a kernel size of 1×1, which can change the number of channels and uniformly resize channels to 256. Then, the high-layer feature maps are processed via nearest interpolation by a factor of 2 for up-sampling, following which it is fed into the co-ordinate attention block to produce attention weights for merging the feature maps. Finally, to ensure the stability of the extracted features, an additional convolutional operation with a kernel size of 3 was added after feature fusion. The processing of the CAFPN can be described as per Equation (Equation 2):(2)$$Y_{i}=C_{i}\left(X_{i}\right)+F_{u}^{\prime }\left(Y_{i+1}\right)$$
where, $Y_{i}$ represents the fused feature maps of the i-th in the FPN, $Y_{i+1}$ denotes the feature maps of (i+1)-th layers from FPN, and i=3,2,1 as $F_{u}^{\prime }$ denote the processed feature maps from the co-ordinate attention blocks, which are introduced as follows.

**Co-ordinate attention blocks:** In this work, we have designed the co-ordinate attention block to capture not only the cross-channel but also direction-aware and position-sensitive pest features, which helps the detection model to more accurately locate the pest targets. Figure 5b presents the architecture of the co-ordinate attention block. Firstly, we begin by decomposing the global pooling operation to yield a pair of direction-aware feature maps, enhancing the network’s ability to accurately localize the agricultural pest. To be specific, it first utilizes the spatial pooling kernels (H,1) and (1,W) to encode each channel’s information along the horizontal and vertical co-ordinates; the generated feature map of the *l*-th channel at height *h* and width *w* is expressed by(3)$$z_{l}^{h}\left(h\right)=\frac{1}{W}\sum_{0\leq i\leq W}x_{l}\left(h,i\right)$$(4)$$z_{l}^{h}\left(w\right)=\frac{1}{H}\sum_{0\leq j\leq H}x_{l}\left(j,w\right)$$The above two transformations will generate a pair of direction-aware feature maps, which help the network to more accurately locate the pests of interest. Subsequently, after encoding the precise positional information, we further need to obtain the co-ordinate attention generation. Initially, the feature maps $z_{c}^{h}\left(h\right)$ and $z_{c}^{w}\left(w\right)$ are concatenated, followed by processing using a convolutional layer for transformation, as represented by(5)$$f=\phi (conv\left(\left[z^{h},z^{w}\right]\right))$$Here, [,] denotes spatial dimension concatenation, and ϕ refers to the nonlinear activation function. The resulting feature map *f* encodes spatial information in both horizontal and vertical directions.

Next, the obtained feature map *f* is divided into two separate tensors, $f^{h}$ and $f^{w}$, and two distinct 1×1 convolutional operations are applied to adjust the number of channels of $f^{h}$ and $f^{w}$ to keep pace with that of the input feature map *X*, which can be depicted by(6)$$g^{h}=\varphi \left(conv_{h}\left(f^{h}\right)\right)$$(7)$$g^{w}=\varphi \left(conv_{h}\left(f^{h}\right)\right)$$

Here, φ(·) signifies the sigmoid function, where both $conv_{h}$ and $conv_{w}$ are the 1×1 convolutional operation. Finally, the attention weights $g^{h}$ and $g^{w}$ are combined with the input feature map, and the output feature maps, *O*, with co-ordinate attention are produced, as defined by(8)$$o\left(i,j\right)=x\left(i,j\right)\times g^{h}\left(i\right)\times g^{w}\left(j\right)$$
where *i* and *j* represent location information. The output feature map contains precise co-ordinate information, aiding in the accurate localization of pests and consequently improving recognition capabilities.

#### 2.2.3. Multi-Class Pest Detection Head

In order to accomplish the detection of multiple crop pest classes, the anchor-free detection framework was adopted because it has the following advantages: (1) eliminating the need for manual hyper-parameter tuning in anchor settings; (2) simplifying the network structure of the detection head; and (3) reducing training memory requirements; therefore, in our work, an anchor-free detection head was adopted as the pest detection head, and this was incorporated with the single-stage detection module. As illustrated in Figure 5c, the multi-class pest detection head contains three subnetworks for predicting category, localization, and the center point of the bounding box of a pest instance. To be specific, the classification subnet outputs a C-dimensional vector (C is the number of categories of AgriPest21); the localization subset outputs the class-specific distances from the center point to the boundaries of the bounding box. The centerness branch generates one-dimensional outputs for predicting the center point of the bounding box. In this study, the two subnetworks, including localization and center point prediction, share the same feature map.

#### 2.2.4. Dynamic Sample Selection Strategy

For the learning of our designed network, some training samples, including positive and negative samples, need to be selected. The original IoU-based sample selection method tends to assign negative labels for small-pest instances, causing small-scale pests to be ignored during training. In this paper, by taking the prior center into consideration, a category-specific Gaussian-shaped function, termed G, is introduced for positive weighting. Each distinct category possesses its own unique set of learnable parameters, whereas pests belonging to the same category share a common set. In this study, we formulate G as follows:(9)$$G\left(d|\mu ,\theta \right)=e^{\frac{-(d-\mu )}{2\theta^{2}}}$$

Here, *d* signifies the displacements from a particular position of a pest instance to its box center, encompassing both the x and y axes and accommodating negative values. μ and θ are learnable parameters of shape (C,2), where *C* denotes the number of categories in the AgriPest21 dataset (C = 21 in this study). These parameters are optimized through back-propagation. The parameter μ is used to control the offset of the center of each class of agricultural pest, and θ evaluates the importance of each position based on category characteristics. Thus, θ decides the number of point samples that effectively contribute to positive loss.

Intuitively, the point sample within the bounding box with more accurate predictions should be paid more attention. However, the network parameters are randomly initialized at the beginning, resulting in unreasonable predicted classification scores. Therefore, guidance information from prior knowledge is extremely important. For a point sample, $i\in S_{n}$, the category-specific prior $G(d_{i})$ from the center weighting module is adopted to produce the positive weights as(10)$$w_{i}^{+}=\frac{G(d_{i})}{\sum_{j\in S_{n}}G\left(d_{i}\right)}$$

A large amount of background area will lead to false detection; thus, it requires a negative weight to suppress the influence of negative samples. In addition, it is known that a point sample within the bounding box usually has high classification confidence, which can be adopted so as to produce an unbiased indicator of false positives. Unfortunately, the negative classification does not provide a gradient for the regression task, implying that the localization confidence should not be further optimized. Therefore, we utilize the values of intersection over union (IoU) between each predicted bounding box of the point sample and the bounding box of all objects to produce the negative weights as(11)$$w_{i}^{-}=1-N\left(\frac{1}{1-k_{i}}\right)$$Here, $k_{i}$ denotes the maximum IoU between the bounding box of location $i\in S_{n}$ and the ground truth boxes. The function *N* is used to normalize the generated negative weights. This causes sharpness in the weight distributions, guaranteeing that the area with the highest IoU incurs no penalty from negative loss. For all the point samples that lie outside the bounding boxes, $w_{i}^{-}$ is assigned a value of 1 since they are definitively background regions.

Finally, based on the assignment results of each sample, we introduced a loss function with weights to optimize the built crop pest detection module. The loss function can be defined by(12)$$L_{cls}=-w^{+}\times ln\left(p\right)-w^{-}\times ln\left(1-p\right)$$(13)$$L_{reg}=w^{+}\times l_{reg}\left(b,b^{*}\right)$$

Here, $w^{+}$ and $w^{-}$ are the negative and positive weights, respectively, while *p* represents the classification confidence. *b* and $b^{*}$ denote the predicted box and ground truth (GT) bounding box, respectively. Generally, $w^{+},w^{-}\in 0,1$, and $w^{+}+w^{-}=1$. However, from our explorations, we found that the prior distribution plays a crucial role in label assignment, particularly during the early training stage. Typically, the distribution of agricultural pests adheres to a prior center. The objects of distinct categories may exhibit different distributions. Simply sampling center positions fails to capture the varying distributions of real-world instances. Ideally, adaptive center distributions are preferred for objects of different categories.

## 3. Experiments

This section introduces the experimental settings, including the parameter settings of the modules, the comparison of detection methods, and the verification metrics. Then, the experimental results of the proposed method and compared detection methods are shown and analyzed. Additionally, the results of the ablation experiments, which explore the effects of the CAFPN and dynamic sample strategy, are shown.

### 3.1. Experimental Setup

**Training parameters:** Similar to FCOS, we initialized the detection head. ResNet50, ResNet101, and Swin Transformer were chosen as the backbones to extract the base features of pests. Notably, we employed the stochastic gradient descent (SGD) optimizer with a momentum of 0.9 to train the network. The training process spanned 24 epochs with an initial learning rate set at 0.0025 and a mini-batch size of 2. It is worth mentioning that the learning rate was decreased by a factor of 10 at the 16th and 19th epochs. All detection models were constructed based on MMDetection using a single NVIDIA TITAN GPU with 24 GB of memory.

**Comparison methods:** The detectors, like Adaptive Training Sample Selection (ATSS) [29], Feature Selection Anchor-free detector (FSAF) [30], FCOS [9], and FreeAnchor [8], were adopted as the compared methods. The performance of our method was evaluated on the AgriPest21 dataset, which we constructed, and this was juxtaposed against other state-of-the-art single-stage detectors. Notably, we kept the parameters of the compared detectors unchanged unless specified otherwise, ensuring the integrity of our experimental findings.

**Evaluation metrics:** In order to conduct a comprehensive and unbiased evaluation of the detectors, we employed the evaluation metrics outlined in [31]: mean average precision and average recall (AR). To be specific, the metric mAP includes $AP^{0.5}$, $AP^{0.75}$, and $AP^{\left[0.5:0.95\right]}$, which are computed across various IoU thresholds. Meanwhile, we also examined $AP^{s}$, $AP^{m}$, and $AP^{l}$ to assess the detection accuracy of small, medium, and large pests, respectively. Here, the scales are defined as follows: small (area: <$32^{2}$ pixels), medium ($32^{2}$ pixels ≤ area: <$96^{2}$ pixels), and large (area: ≥$96^{2}$ pixels).

### 3.2. Pest Detection Results

**The performance of each type of agricultural pest on AgriPest21 dataset:** We conducted various experiments on the AgriPest21 dataset. Table 2 demonstrates the detection results for each category of agricultural pest. It shows that the average mAP of 21 types of agricultural pests is 77.2% for our method, which outperformed other detection modules. We also note that the detection AP of “HoF” is much lower than that of other categories of pests, which is primarily attributed to the limited number of samples available for this class, hindering an effective understanding of its unique characteristics. However, when compared to other methods, our method can still obtain 25.2% for AP for the HoF pest, which is better than other methods.

The performance on IP102 dataset: To further demonstrate the effectiveness of the proposed method for detecting pests with various scales and under complex backgrounds, we have obtained experimental results on the IP102 dataset, as shown in Table 3. It shows the detection performance under different IoUs and scales. The proposed method obtains 29.8% and 49.7% when the IoUs are set to [0.5:0.95] and 0.5. Concurrently, the mAP values are 8.9%, 27.9%, and 29.3% for small, medium, and large pests. These detection results outperform other detectors. However, we also note that, compared with the results on the AgriPest21 dataset, the accuracy of the detection is relatively low compared to AgriPest21 dataset. This is due to the serious problem of sample imbalance, namely the long tail distribution problem.

**Experimental results using different detectors:** To demonstrate the performance of the proposed method, i.e., CAFPN and dynamic sample selection strategy, we have merged these developed methods with other detection modules, as shown in Table 4. The introduction of these methods significantly improves the detection accuracy of agricultural pests. Thus, these experimental results demonstrate that the proposed two modules are valid.

**Performance of the proposed method under different IoUs:** As we know, IoU is used to verify the quality of pest localization. The larger the IOU, the higher the localization accuracy. Therefore, in order to evaluate the localization accuracy of the detection results, Table 5 shows the results under different IoUs. It demonstrates that the proposed approach obtains 77.2% and 62.7% for mAP when the IoUs are set to 0.5 and 0.75, respectively. We also calculated the mAP when the IoU was set from 0.5 to 0.95 with an interval of 0.05. Our method can obtain 51.6% for $AP^{\left[0.5,0.95\right]}$, which obviously outperforms the other detectors. These results suggest that our method can improve agricultural pest accuracy recognition.

**Performance at different scales of agricultural pests:** We also illustrate the performance of our method for detecting crop pests with different scales, especially small scale, as shown in Table 6. It shows that our method achieves superior performance on the AP and AR for small, medium, and large pests. Especially, for small-pest detection, our method demonstrates superior performance, outperforming other detection modules, with an AP that is 4.8% higher than the second-best method (ATSS). These improvements contribute to the powerful ability of CAFPN in detecting small pests.

### 3.3. Experimental Results with Different Backbones

In this study, the commonly used ReseNet 50 was selected as the backbone to extract the basis features of agricultural pests. However, we may have to question whether a deeper network might improve pest detection accuracy. Therefore, we explored the performance of detectors that used deeper backbone networks, such as ResNet101 and Swin Transformer. Table 7 demonstrates the detection results of different modules with different backbones. It suggests that deeper backbones only slightly improved detection accuracy in terms of the AP evaluation metric. We also observe that a deeper network is harmful to the detection of small pests. For example, the FCOS detection module using ResNet101 achieves 28.3% for AP for small objects, which is lower than when using ResNet50 as the backbone. This is because, with the increase in network depth, the information on small-scale pests is severely lost, leading to poor performance for small-pest detection. Furthermore, the increase in network depth also increases the time needed for network training and testing. Thus, in this study, we selected ResNet50 as the backbone to achieve optimal detection performance.

### 3.4. Efficiency Analysis

We are aware that detection efficiency is also an important factor in the task of crop pest detection. Therefore, in order to assess the detection efficiency of our proposed network, we meticulously evaluated key metrics such as number of parameters and the testing speed (frames per second, FPS) of our module, benchmarking it against other comparable methods. From Table 8, we can observe that the GFOLPs, the number of parameters, and the detection speed of our module are 211.37, 32.19M, and 25.8FPS, respectively, showing that it outperforms most of the compared methods and is slightly lower than the single-stage anchor-free module, FCOS. However, the detection accuracy of the proposed method remarkably surpassed the FCOS detector.

In order to further analyze the impact of the CAFPN and the dynamic sample selection strategy on model efficiency, we conducted an ablation experiment, as shown in Table 9. This shows that the CAFPN can increase the number of parameters from 32.02 M to 32.19 M, and the addition of the dynamic sample selection strategy has no impact on efficiency. In summary, the proposed method achieves a significant improvement in detection accuracy with a lower computational burden.

### 3.5. Ablation Experiment

We conducted a series of experiments to investigate the impact of the CAFPN and the dynamic sample selection strategy. The detection outcomes on the AgriPest21 dataset are summarized in Table 10. For our baseline, we utilized the FCOS detector paired with a ResNet50 backbone. Notably, when we integrated the proposed CAFPN in lieu of the traditional FPN, the AP increased to 49.3%, indicating that the co-ordination attentional feature fusion pyramid network contributed significantly to enhancing pest detection. Specifically, the introduction of the CAFPN boosts the accuracy of detecting small-scale pests. Furthermore, incorporating the dynamic sample selection strategy into the training process elevated performance to 51.6%, achieving a 2.3% improvement. This underscores the crucial role of the dynamic sample selection strategy in network training.

### 3.6. Quantitative Examples

In Figure 6, we visualize some of the detection results when using the AgriPest21 dataset to illustrate the performance of the proposed method. It shows that our method is also efficient in the detection of tiny-sized pests in a dense distribution. To be specific, on the left of Figure 6, we can see that our method can recognize and detect pests at tiny scales, and, for the unlabeled pest instances, our method can still detect these. At the same time, in the middle and on the right of Figure 6, it is evident that our method can achieve precise detection for small-sized and densely distributed pests. We provide some detection results on the IP102 dataset, as shown in Figure 7. From Figure 7d, we can see that the "grub" pest cannot be detected. This is due to the posture and shape of the pest, which affects the extraction of its features, resulting in poor detection performance. Moreover, the pest “Asiatic rice borer” is classified as “yellow rice borer” in Figure 7e, and the pest “brown crop hopper” is mistaken as “small brown crop hopper” in Figure 7g. This may because of the high similarity among these types of pests.

## 4. Conclusions

This study developed a co-ordinate-attention-based pyramid network that captures cross-channel, direction-aware, and position-sensitive features in digital images to improve the accurate detection of small agricultural pests. In addition, to train the proposed pest detection network, a dynamic positive and negative sample selection strategy was introduced to select positive and negative training samples. By combining this with a single-stage anchor-free detection module, FCOS, we accomplished the accurate detection of agricultural pests. Extensive and thorough experiments were conducted on the extensive AgriPest21 dataset, clearly establishing that our proposed approach yields outstanding results and surpasses other detection methods in both precision and efficiency. Additionally, the ablation studies further corroborate the efficacy of each individual component we have introduced. Nevertheless, the experimental findings also reveal that our proposed method still possesses certain limitations. According to the experimental outcomes, our proposed method exhibits certain constraints; for instance, the module has poor performance in few-shot pest detection, as mentioned in Section 3.1. Thus, in future work, more focus will be directed to pest detection with fewer examples by adopting data generation strategies, such as diffusion models, generative adversarial networks, and other state-of-the-art methods.

## Figures and Tables

**Figure 1 insects-16-00103-f001:**
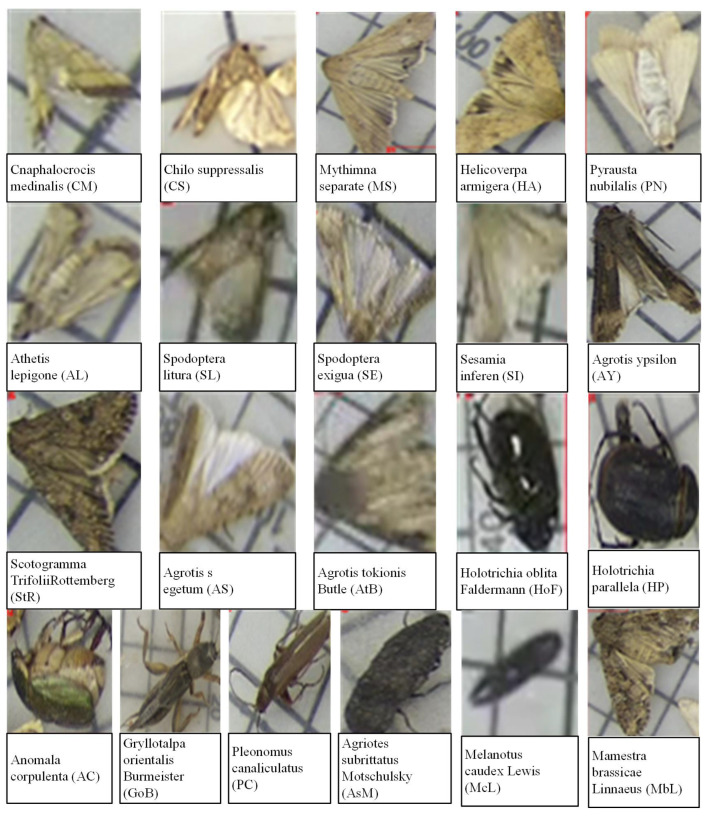
The images of each type of agricultural pest.

**Figure 2 insects-16-00103-f002:**
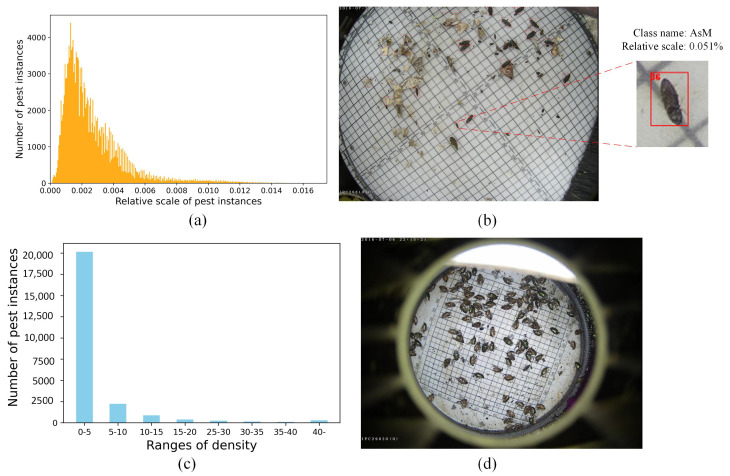
The distribution of the agricultural pest dataset. The relative scale of pest instances (**a**) and an example of pest image with a relative scale of 0.051% (**b**); the density distribution of crop pest dataset (**c**) and an example of pest image with dense distribution (**d**).

**Figure 3 insects-16-00103-f003:**
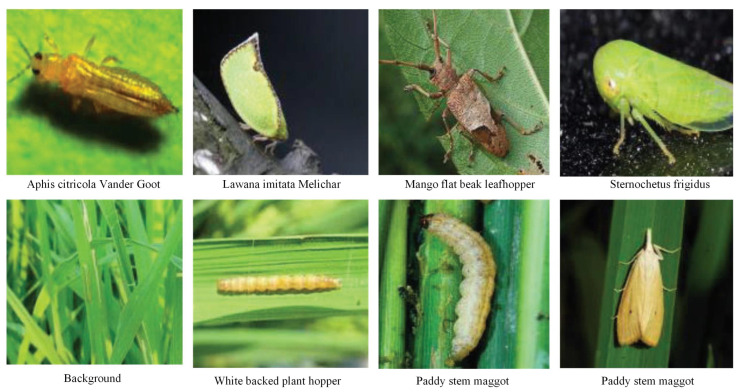
Some examples of the IP102 dataset.

**Figure 4 insects-16-00103-f004:**
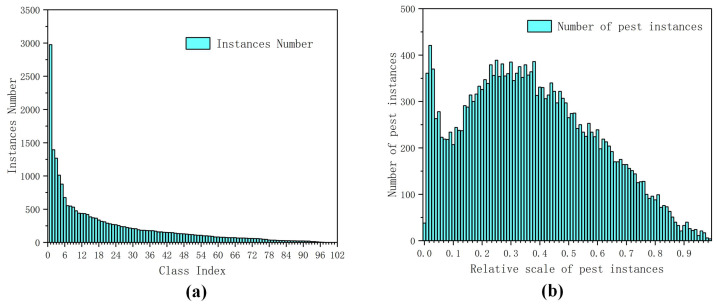
Analysis of the IP102 dataset: (**a**) is the distribution of number of pest instances per type of pest; (**b**) is the distribution of relative scales of pest instances.

**Figure 5 insects-16-00103-f005:**
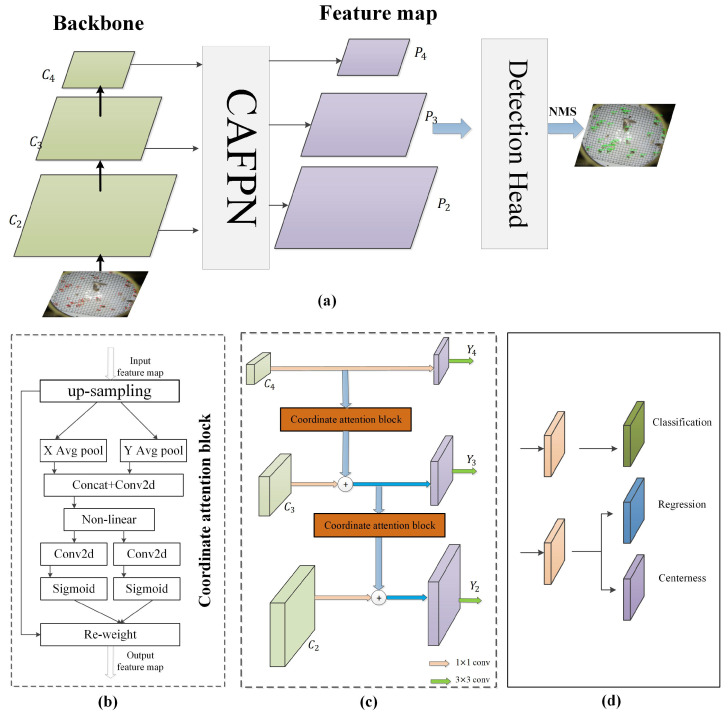
Overall framework of the agricultural pest detection module (**a**); (**b**) is the architecture of the proposed co-ordinate attention block; (**c**) is the co-ordinate-attention-based feature pyramid network; (**d**) is the crop pest detection head.

**Figure 6 insects-16-00103-f006:**
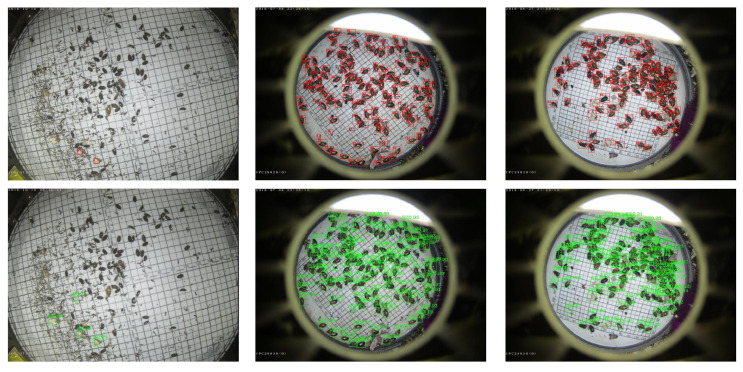
Some examples from the small-pest dataset AgriPest21; the red and green boxes represent the annotated and predicted bounding boxes, respectively.

**Figure 7 insects-16-00103-f007:**
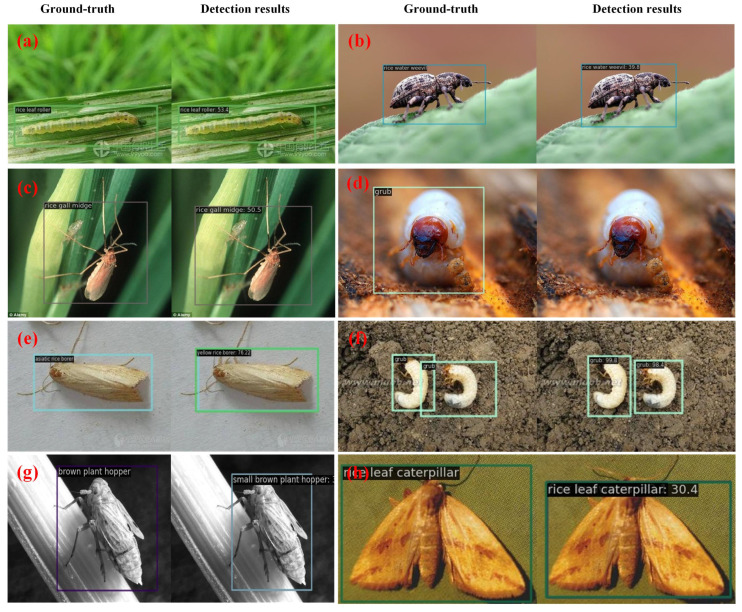
Visualization of detection results on IP102 dataset. (**a**–**h**) represent the detection results of the proposed method for rice leaf roller, rice water weevil, rice gall midge, grub, asiatic rice borer, grub, brown plant hopper, and rice leaf caterpillar. The left of each sub-figure denotes the ground truth and the right of the sub-figure is the detection result.

**Table 1 insects-16-00103-t001:** Details of the 21 types of agricultural pests.

Classes	Training Dataset	Testing Dataset	Average Relative Scale (%)
	Number of Instances	Number of Pest Images	
*Cnaphalocrocis medinalis* (CM)	1224	932	0.1214
*Chilo suppressalis* (CS)	1285	454	0.1793
*Mythimna separate* (MS)	8374	3637	0.3978
*Helicoverpa armigera* (HA)	26,588	8740	0.2814
*Pyrausta nubilalis* (PN)	15,739	5294	0.2267
*Athetis lepigone* (AL)	28,932	7200	0.1298
Spodoptera litura (SL)	1896	1543	0.4572
*Spodoptera exigua* (SE)	7116	3527	0.1377
*Sesamia inferen* (SI)	1768	1335	0.2776
*Agrotis ypsilon* (AY)	3890	2314	0.5703
*Mamestra brassicae* Linnaeus (MbL)	2170	1632	0.4259
*Scotogramma trifolii* Rottemberg (StR)	4393	3051	0.2816
*Agrotis segetum* (AS)	1615	1330	0.4024
*Agrotis tokionis* Butle (AtB)	465	351	0.6375
*Holotrichia oblita* Faldermann (HoF)	82	70	0.3348
*Holotrichia parallela* (HP)	11,325	3002	0.2518
*Anomala corpulenta* (AC)	52,134	5083	0.2466
*Gryllotalpa orientalis* Burmeister (GoB)	6480	3589	0.9530
*Pleonomus canaliculatus* (PC)	157	109	0.3281
*Agriotes subrittatus* Motschulsky (AsM)	6161	1729	0.1129
*Melanotus caudex* Lewis (McL)	677	224	0.1584

**Table 2 insects-16-00103-t002:** Detection results (AP) of each type of pest on AgriPest21 datasets (unit: %).

Methods	RetinaNet	ATSS	FSAF	FCOS	FreeAnchor	Our Method
Class						
CM	68.2	73	72.8	62.8	72.1	74.7
CS	73.1	79.1	77.9	69.9	74.9	81.7
MS	75.3	81.6	81.9	81	79.3	84.9
HA	88.1	90.8	91.2	90.4	88.3	91.6
PN	76.7	82.6	82.7	80	79.5	84.9
AL	62.8	76.3	76	73.1	64.2	77.6
SL	78.3	82.5	80.8	83.4	83.2	84
SE	48.1	56.2	53.6	49.5	51.9	60.7
SI	79.1	84.4	80.5	77.8	79.9	85.7
AY	83.7	88.7	88.3	87.1	87.5	90.2
MbL	57.6	64.5	61.5	63.3	62.7	75.4
StR	52.5	59	55.1	55.3	55.2	65.9
AS	42.5	54.4	52.3	48.4	46.8	60.5
AtB	44.7	58.6	48.8	50.1	48.3	59.3
HoF	7.3	8	3.3	10.5	8	25.2
HP	87.8	92.1	91.6	90.4	87.4	91.8
AC	89.3	90.6	90.6	90.6	87.8	91.5
GoB	98.2	98.5	98.5	98.2	98.2	98.1
PC	43.4	55.1	52.6	54.8	51.6	61.5
AsM	75.2	82.7	83	80.3	77.9	85.6
McL	27.6	72.5	74	68.7	51.7	89.8
Average	64.7	72.9	71.3	69.8	68.4	77.2

**Table 3 insects-16-00103-t003:** Experimental results on the IP102 dataset.

Methods	AP[0.5,0.95]	AP0.5	AP0.75	APs	APm	APl	ARs	ARm	ARl
RetinaNet	25.9	44.8	26.7	4.2	25.7	27.1	6.8	45.6	55.4
ATSS	21.2	39.1	20.3	7.8	22	22.2	10.5	42.4	53.6
FASF	23.9	44.8	22.8	8	27.9	25.1	6	44.3	51.2
FCOS	11.5	22.9	10.1	2.8	15.3	12.1	5.9	38.6	49.9
FreeAnchor	26.9	46.1	27.7	3.4	27.6	28	4.9	46.9	56
Our method	29.8	49.7	29.2	8.9	27.9	29.3	11.6	49.1	59.4

**Table 4 insects-16-00103-t004:** Experimental results for different detectors on AgriPest21 dataset.

Methods	CAFPN	Dynamic Sample Selection Strategy	AP	AP0.5	AP0.75
RetinaNet	✓	×	46.5	67.2	55.8
	✓	✓	48.9	69.3	61.3
ATSS	✓	×	48.2	73.1	54.5
	✓	✓	50.2	75.8	59.9
FASF	✓	×	47.4	73.6	57.9
	✓	✓	49.6	75.8	60.2
FreeAnchor	✓	×	47.2	73.2	57.8
	✓	✓	49.6	74.2	61.1
FCOS	✓	×	49.3	75.3	58.9
	✓	✓	51.6	77.2	62.7

**Table 5 insects-16-00103-t005:** The experimental results under different IoUs (unit: %).

Methods	AP[0.5,0.95]	AP0.5	AP0.75
RetinaNet	41.3	65.1	48
ATSS	46.4	72.6	51.4
FSAF	44.4	70	52.3
FCOS	44.2	69.2	51.4
FreeAnchor	43.6	67.9	51.3
Our method	51.6	77.2	62.7

**Table 6 insects-16-00103-t006:** Performance comparison for the tiny pest dataset AgriPest21 (unit: %).

Methods	APs	APm	APl	ARs	ARm	ARl
RetinaNet	25.1	47.3	50	42.3	65.9	50
ATSS	32	51.5	40	52.3	64.2	40
FSAF	29.3	49.3	40	52.3	64.2	40
FCOS	29.6	49.1	40.1	48.1	61.4	40
FreeAnchor	28.1	48.9	35	47.3	65.4	35
Our method	36.8	57.6	69.5	51.6	67.3	55

**Table 7 insects-16-00103-t007:** Experimental results using different backbones (unit: %).

Method	Backbones	AR	AP	APs
ATSS	ResNet50	63.8	46.4	32.2
	ResNet101	63.3	46.8	32.8
RetinaNet	ResNet50	59.7	41.3	25.1
	ResNet101	60.4	41.8	25.7
FSAF	ResNet50	63.2	44.4	29.3
	ResNet101	63.2	45	30.6
FCOS	ResNet50	60	44.2	31
	ResNet101	60	44.5	28.3
Free-Anchor	ResNet50	60.9	43.6	26.9
	ResNet101	60.9	43.2	26.6
Our method	ResNet50	69.3	51.6	36.8
	ResNet101	69.3	51.2	36.4
	Swin Transformer	69.8	51.4	36.7

**Table 8 insects-16-00103-t008:** The detection efficiency of the proposed pest detection module and comparison methods.

Method	GFlOPs	Parameters (M)	Speed (FPS)
ATSS	215.58	32.07	22.6
RetinaNet	223.89	36.52	21.8
FSAF	231.56	36.06	21.1
FCOS	209.75	32.02	26.2
FreeAnchor	251.31	37.74	20.2
Our method	211.37	32.19	25.8

**Table 9 insects-16-00103-t009:** Efficiency analysis of the proposed model when using CAFPN and the soft-label-assignment strategy (unit: %).

Backbone	CAFPN	Dynamic Sample Selection Strategy	Parameter (M)	Detection Speed (FPS)
	×	×	32.02	26.2
ResNet50	✓	×	32.19	25.8
	✓	✓	32.19	25.8

**Table 10 insects-16-00103-t010:** The influence of CAFPN and the Dynamic sample selection strategy (AgriPest21 dataset) (unit: %).

	CAFPN	Dynamic Sample Selection Strategy	AP	AP0.5	AP0.75	APs	APm	APl
	×	×	44.2	69.2	51.4	29.6	49.1	40.1
FCOS	✓	×	49.3	75.3	58.9	34.5	53.5	50.1
	✓	✓	51.6	77.2	62.7	36.8	57.6	69.5

## Data Availability

Due to the confidentiality agreement, this dataset cannot be fully disclosed temporarily. However, if there is a research need, the researchers can contact the author via email to obtain it.

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
