# Peer review of "Precise Crop Pest Detection Based on Co-Ordinate-Attention-Based Feature Pyramid Module"

_insects, 2025, doi:10.3390/insects16010103_

Round 1
Reviewer 1 Report (Previous Reviewer 1)
Comments and Suggestions for Authors
In this paper, the authors proposed a new crop pest detection framework, including an attention-based multiscale feature pyramid network for pest feature extraction of small pests, and a dynamic sample-selection strategy for selection of positive and negative samples during training. The core idea is simple and interesting. These proposed methods are also effective and outperform other sota modules. A large number of experiments are conducted on two crop dataset, including AgriPest21 and IP102 dataset. However, the results on IP102 dataset should be added in Abstract. In figure 7, the titles of subfigures should be given. And after minor revision, the paper can be accepted.
Author Response
General comments:
In this paper, the authors proposed a new crop pest detection framework, including an attention-based multiscale feature pyramid network for pest feature extraction of small pests, and a dynamic sample-selection strategy for selection of positive and negative samples during training. The core idea is simple and interesting. These proposed methods are also effective and outperform other sota modules. A large number of experiments are conducted on two crop dataset, including AgriPest21 and IP102 dataset. However, the results on IP102 dataset should be added in Abstract. In figure 7, the titles of subfigures should be given. And after minor revision, the paper can be accepted.
Answer: Thank you very much for your affirmation and suggestions for our work. As your suggestions, we have improved the manuscript.
Specific comments:
Question 1: However, the results on IP102 dataset should be added in Abstract.
Answer 1: Thanks for your advice. We have added the experimental results on IP102 dataset in the part of Abstract. More details are shown in the revised manuscript.
Question 2: In figure 7, the titles of subfigures should be given.
Answer 2: Thanks for your suggestion. We have revised the title of Figure 7, and added the title of each sub-figure, as shown in the revised manuscript.
Finally, we tried our best to improve the manuscript and made some changes in the manuscript. And here we did not list the changes but marked in red in the revised paper. We appreciate for Editors’ and Reviewers’ warm work earnestly and hope that the correction will meet with approval. We look forward to hearing from you regarding our submission. We would be glad to respond to any further questions and comments that you may have.

Reviewer 2 Report (Previous Reviewer 2)
Comments and Suggestions for Authors
The improvements added to the previous version are acceptable.
in this "new" version, Tables 9, 10 and 8 do not appear in ascending order. You must follow the format rules.
Author Response
Thank you for your letter and reviewers’ comments concerning our manuscript entitled “Precise Crop Pest Detection based on Co-ordinate Attention-Based Feature Pyramid Module” (ID: insects-3415829) which is submitted on the journal of Insects. Those comments are valuable and very helpful for revising and improving our paper, as well as the important guiding significance to our researchers. We have studied comments carefully and made corrections which we hope meet with approval. Revised portions are marked in red in the revised paper. The main corrections in this paper and the response to reviewer’s comments are listed as following:
Comments:
The improvements added to the previous version are acceptable.
In this "new" version, Tables 9, 10 and 8 do not appear in ascending order. You must follow the format rules.
Answer: Thanks so much for your suggestion. We have revised and edited the layout of the manuscript, and the Table 8,9,10, and Tables 8, 9, and 10 are arranged in order.

This manuscript is a resubmission of an earlier submission. The following is a list of the peer review reports and author responses from that submission.
Round 1
Reviewer 1 Report
Comments and Suggestions for Authors
Please find the attachted file.

Author Response
General comments:
In this paper, to address the precise detection of 21 types of agricultural pests, the authors design a coordinate attention-based feature pyramid network, which can capture not only the cross-channel but also direction-aware and position-sensitive pest features. Also, for network training, a soft label assignment strategy is introduced to select positive and negative training samples by considering the high classification score and the precise localization. Finally, numerous comprehensive experiments on large-scale AgriPest21 datasets demonstrate that the proposed approach achieves excellent results and outperforms state-of-the-art detection methods in terms of accuracy and efficiency. Overall, the quality of this manuscript is good. The research on precise detection of agricultural pests is very meaningful, and the proposed module is valuable. However, a minor revision is required for enhancing the quality of the manuscript.
Answer: Thank you very much for your affirmation and suggestions for our work. As your suggestions, we have improved the manuscript.
Specific comments:
Question 1: Some important references related to attention mechanism are missing.
Answer 1: Thanks for your advice. As your comment, we have added some introduction about attention mechanism. More details are shown in the revised manuscript.
Question 2: This work is laudable and of interest to the research community. For the development of the field, it is desirable that this dataset be published as a benchmark.
Answer 2: Thanks. Sorry, due to the confidentiality agreement, this dataset cannot be made public temporarily. However, if there is a research need, please contact the author via email to obtain it.
Question 3: In Line 261, 12 should be revised as 21.
Answer 3: Thanks. We are so sorry for our neglect. We have revise the mistake.
Question 4: Please explain why the coordinate attention mechanism is used in this manuscript?
Answer 4: Thanks for your comments. It is known that the coordinate attention is the direction-aware and position-sensitive. As we know, the feature fusion mechanism within an FPN simply involves adding the up-sampled higher-level feature maps. This approach ignores the positional information inherent in the feature maps, which holds significance in producing spatial attention weight maps. Therefore, a novel attention mechanism related to position information was developed, which can improve the module’s ability for feature extraction.
Question 5: (5) The Equation (1) misses a “)”..
Answer 5: We are so sorry for our ignorance We have revise the error. Thanks.
Question 6: In Equation (4), “0 < i < w” should be “0 < i < h”?
Answer 6:We have corrected it. Thanks.
Question 7: Please give the description of Figure 2.
Answer 7:Thanks so much for your valuable advice. We have added some description of Figure 2, as shown in the revised manuscript.
Finally, we tried our best to improve the manuscript and made some changes in the manuscript. And here we did not list the changes but marked in red in the revised paper. We appreciate for Editors’ and Reviewers’ warm work earnestly and hope that the correction will meet with approval. We look forward to hearing from you regarding our submission. We would be glad to respond to any further questions and comments that you may have.

Reviewer 2 Report
Comments and Suggestions for Authors
Authors designed a co-ordinate attention-based feature pyramid network to to extract the rich features of small pests. Also, they introduce a dynamic sample-selection strategy using positive and negative weight functions. After several conducted experiments an average accuracy of 77.2% has been achieved, witch is higher than several other related works compared in this study.
Section 3.3. is confusing. Table 7 and 6 are disordered in text, what makes also difficult to understand the text. Must be confirmed and improved.
Author Response
Thank you for your letter and reviewers’ comments concerning our manuscript entitled “Precise Agricultural Pest Detection via a Co-ordinate Attention-Based Feature Pyramid Module” (ID: insects-3219756) which is submitted on the journal of Insects. Those comments are valuable and very helpful for revising and improving our paper, as well as the important guiding significance to our researchers. We have studied comments carefully and made corrections which we hope meet with approval. Revised portions are marked in red in the revised paper. The main corrections in this paper and the response to reviewer’s comments are listed as following:
Responds to reviewer’s comments:
Reviewer #2:
Comments: Authors designed a co-ordinate attention-based feature pyramid network to extract the rich features of small pests. Also, they introduce a dynamic sample-selection strategy using positive and negative weight functions. After several conducted experiments an average accuracy of 77.2% has been achieved, which is higher than several other related works compared in this study.
Section 3.3. is confusing. Table 7 and 6 are disordered in text, what makes also difficult to understand the text. Must be confirmed and improved.
Answer: Thanks for your suggestion. We are so sorry for confusing you. We correct the order of Table 6 and Table 7. Besides, we have added the experimental results of detection efficiency of the proposed pest detection module and comparison methods. In addition, the order of original Table 6 and Table 7 have been correct to Table 7 and Table 8.
Finally, we tried our best to improve the manuscript and made some changes in the manuscript. And here we did not list the changes but marked in red in the revised paper. We appreciate for Editors’ and Reviewers’ warm work earnestly and hope that the correction will meet with approval. We look forward to hearing from you regarding our submission. We would be glad to respond to any further questions and comments that you may have.

Reviewer 3 Report
Comments and Suggestions for Authors
This paper proposes a Coordinate Attention-Based Feature Pyramid Network, CAFPN, for enhancing feature extraction of small pests. It also suggests a dynamic sample selection strategy to optimize training by balancing the weights of positive and negative samples. The proposed scheme achieved 77.2% mAP detection accuracy in large-scale experiments on a diverse dataset of crop pests.
I have a few recommendations to improve the quality of the manuscript.
The descriptions of the 'Coordinate Attention-based Feature Pyramid Network' and dynamic sample selection strategy are worded in a somewhat confusing manner and are not clearly expressed. This points out the importance of clarity and conciseness in research; such might make it hard for any researcher to replicate the study or apply the methodology to other datasets and conditions.
Because most of the results are based on a single dataset, understanding the proposed model's possible performance in different agricultural environments, as well as with other crop types and their respective pest species, is limited.
Is the dataset used for training and testing the model publicly available? If not, then it reduces the ability of the general research community to reproduce or further advance the findings.
While the primary evaluation metric of interest seems to be the mean Average Precision, other metrics such as recall, precision, F1-score, and computational efficiency in operational conditions would give further insights into the performance of the model.
The paper compares the proposed method with other detectors, but it does not really present a comprehensive comparative analysis, including recent state-of-the-art techniques in the field. More comparisons with qualitative and quantitative features would give merit to the argument of why this proposed model is the best.
While the manuscript mentions gains in the detection of small-sized pests, it still needs to discuss how the model will behave under a variety of pest sizes and densities—a necessary basis for practical applications.
Author Response
Dear Editors and Reviewers:
Thank you for your letter and reviewers’ comments concerning our manuscript entitled “Precise Agricultural Pest Detection via a Co-ordinate Attention-Based Feature Pyramid Module” (ID: insects-3219756) which is submitted on the journal of Insects. Those comments are valuable and very helpful for revising and improving our paper, as well as the important guiding significance to our researchers. We have studied comments carefully and made corrections which we hope meet with approval. Revised portions are marked in red in the revised paper. The main corrections in this paper and the response to reviewer’s comments are listed as following:
Responds to reviewer’s comments:
Reviewer #3:
General comments:
This paper proposes a Coordinate Attention-Based Feature Pyramid Network, CAFPN, for enhancing feature extraction of small pests. It also suggests a dynamic sample selection strategy to optimize training by balancing the weights of positive and negative samples. The proposed scheme achieved 77.2% mAP detection accuracy in large-scale experiments on a diverse dataset of crop pests. I have a few recommendations to improve the quality of the manuscript.
Answer: Thank you very much for your affirmation and suggestions for our work. As your suggestions, we have improved the manuscript.
Specific comments:
Question 1: The descriptions of the 'Coordinate Attention-based Feature Pyramid Network' and dynamic sample selection strategy are worded in a somewhat confusing manner and are not clearly expressed. This points out the importance of clarity and conciseness in research; such might make it hard for any researcher to replicate the study or apply the methodology to other datasets and conditions.
Answer 1: Thanks for your suggestion. We are so sorry for confusing you. So, we have revised the description of our method, as shown in the revised manuscript with red marker. As the same time, we will open source the code of this method on the github, which can help researchers to replicate the study or apply the methodology to other datasets.
Question 2: Because most of the results are based on a single dataset, understanding the proposed model's possible performance in different agricultural environments, as well as with other crop types and their respective pest species, is limited.
Answer 2: Thanks for your comments. In this paper, to address the precise detection of crop pest with small size, we proposed xx module, and we evaluated the effectiveness of our method on the large-scale dataset, AgriPest21. The pest instances of this dataset have small sizes and dense distribution. This is why we choose AgriPest21 dataset to verify our module. As your advice, there are some public crop pest and disease dataset, like IP102, AI Challenger 2018, Plant-Village and so on. However, most of them are used for crop disease classification, and the pest instances in public the crop dataset have large sizes, generally, one pest target in an image, which does not meet the acquirements of the detection of small pest. In addition, to illustrate the performance of our method, we compared it with a large number of SOTA method.
Question 3: Is the dataset used for training and testing the model publicly available? If not, then it reduces the ability of the general research community to reproduce or further advance the findings..
Answer 3: Thanks. Due to the confidentiality agreement, this dataset cannot be fully disclosed temporarily. However, if there is a research need, the researchers can contact the author via email to obtain it. And the code of the proposed module will public on github.
Question 4: While the primary evaluation metric of interest seems to be the mean Average Precision, other metrics such as recall, precision, F1-score, and computational efficiency in operational conditions would give further insights into the performance of the model.
Answer 4: Thanks for your comments. In this paper, to evaluate the performance of our method, the metrics, mAP, AR (average recall of 21 types of crop pests), and the number of parameters, GFLOPs and testing speed of module are used to verify the efficiency of our module. To be specify, in Table 2, We report the AP of each type of pest and mean AP of all classes. To demonstrate the performance of localization, we verify performance of the proposed method under different IoUs, as shown in Table 3. In Table 4 and Table 5, the detection results of average recall (AR) are demonstrated, and we also give the results of ARs,ARm, and ARl, which demonstrate the AR for small, medium and large pest instances.
In Table 6, we give the results of GFLOPs, the number of parameters, and testing speed (frame per second, FPS) to illustrate the detection efficiency of the module.
As for the metric, F1-score, it is an indicator used in statistics to measure the accuracy of binary classification models, which is not inappropriate to evaluate the performance of multi-classes object detection module. Thus, we do not adopt this metric.
Thanks again for your comments.
Question 5: The paper compares the proposed method with other detectors, but it does not really present a comprehensive comparative analysis, including recent state-of-the-art techniques in the field. More comparisons with qualitative and quantitative features would give merit to the argument of why this proposed model is the best.
Answer 5: Thanks so much for your suggestion. According to your comments, we further explained and analyzed why our proposed method is effective, as shown in the revised manuscript.
Question 6: While the manuscript mentions gains in the detection of small-sized pests, it still needs to discuss how the model will behave under a variety of pest sizes and densities—a necessary basis for practical applications.
Answer 6:Thanks so much for your advice, which is valuable to improve our manuscript. In this paper, the proposed method aims to address the precise detection of small pest. And the experiments conducted on the large-scale small pest dataset, AgriPest21, demonstrated that the effectiveness of the proposed method. Besides, for some crop pest with medium and large scales, we also give the detection results, as shown in Table 4 and Table 5. However, we neglect to discuss how the model behave under different densities, only present the detection results under high-density distribution, as shown in Figure 4. In our next work, we will collect more pest images with various densities, and test the experimental results under different densities.
Finally, we tried our best to improve the manuscript and made some changes in the manuscript. And here we did not list the changes but marked in red in the revised paper. We appreciate for Editors’ and Reviewers’ warm work earnestly and hope that the correction will meet with approval. We look forward to hearing from you regarding our submission. We would be glad to respond to any further questions and comments that you may have.

Round 2
Reviewer 3 Report
Comments and Suggestions for Authors
Authors should check the existing literature properly; in response to my previous query, "Because most of the results are based on a single dataset, understanding the proposed
model's possible performance in different agricultural environments, as well as with other crop types and their respective pest species, is limited"
Please explain the response in the reviewer response section in detail to my previous query. I think the authors have not addressed this comment in the manuscript properly: "The paper compares the proposed method with other detectors, but it does not really present a comprehensive comparative analysis, including recent state-of-the-art techniques in the field. More comparisons with qualitative and quantitative features would give merit to the argument of why this proposed model is the best. "
The authors have already proposed a similar scheme in the article https://www.mdpi.com/2075-4450/13/11/978. Authors should justify the newness of the current methodology and cite this manuscript.
In response to my previous query, "While the manuscript mentions gains in the detection of small-sized pests, it still needs to discuss how the model will behave under a variety of pest sizes and densities—a necessary basis for practical applications. " The authors should consider other datasets.
Author Response
Dear Editors and Reviewers:
Thank you for your letter and reviewers’ comments concerning our manuscript entitled “Precise Agricultural Pest Detection via a Co-ordinate Attention-Based Feature Pyramid Module” (ID: insects-3219756) which is submitted on the journal of Insects. We are so sorry that we may not understand the question you raised earlier. Thus, we have carefully studied your question and provided the following answers, hoping to meet your requirements.
Question1: Authors should check the existing literature properly; in response to my previous query, "Because most of the results are based on a single dataset, understanding the proposed model's possible performance in different agricultural environments, as well as with other crop types and their respective pest species, is limited"
Answer 1: We are very sorry that we did not understand the question you raised earlier. In order to solve your problem, we conducted relevant comparative experiments on the publicly available large-scale dataset IP102. The experimental results are shown in Table 3. Besides, in Section 2.1, we also give some analysis about the IP102 dataset. According to the analysis, in Figure 1(a), we can observe that there exits serious imbalance of pest classes for IP102 dataset. And there are various scales in IP102 dataset, and the relative scales of the pest targets of tend to large compared with our built AgriPest 21 dataset, as shown in Figure 1(b). Some experiments are conducted on the IP102 dataset to evaluate the performance of method for detecting the pest with various scales or under different backgrounds. This will further verify the generalization of the proposed method. More details are shown in the revise manuscript with red marks.
Figure 1 Analysis of the IP102 dataset, (a) is the distribution of number of pest instances per type of pest; (b) is the distribution of relative scales of pest instances.
Question 2: Please explain the response in the reviewer response section in detail to my previous query. I think the authors have not addressed this comment in the manuscript properly: "The paper compares the proposed method with other detectors, but it does not really present a comprehensive comparative analysis, including recent state-of-the-art techniques in the field. More comparisons with qualitative and quantitative features would give merit to the argument of why this proposed model is the best. "
Answer 2: Thanks for your comments. We are so sorry that we cannot understand your meaning earlier. We have added many experiments to demonstrate the effectiveness of the proposed module, as shown in Table 1.
Table 1 Experimental results of the proposed modules by combining different detectors on AgriPest21 dataset
Methods |
CAFPN |
Dynamic sample selection strategy |
AP |
AP0.5 |
AP0.75 |
RetinaNet |
√ |
× |
46.5 |
67.2 |
55.8 |
√ |
√ |
48.9 |
69.3 |
61.3 |
|
ATSS |
√ |
× |
48.2 |
73.1 |
54.5 |
√ |
√ |
50.2 |
75.8 |
59.9 |
|
FASF |
√ |
× |
47.4 |
73.6 |
57.9 |
√ |
√ |
49.6 |
75.8 |
60.2 |
|
FreeAnchor |
√ |
× |
47.2 |
73.2 |
57.8 |
√ |
√ |
49.6 |
74.2 |
61.1 |
|
FCOS |
√ |
× |
49.3 |
75.3 |
58.9 |
√ |
√ |
51.6 |
77.2 |
62.7 |
Question 3: The authors have already proposed a similar scheme in the article https://www.mdpi.com/2075-4450/13/11/978. Authors should justify the newness of the current methodology and cite this manuscript.
Answer 3: Thanks for your comments. In our previous work, for detecting the corn pest under complex scene, firstly, a deep residual network with deformable convolution has been introduced to obtain the features of the corn pest images; And then, an attention-based multi-scale feature pyramid network has been developed, finally, we combined the proposed modules with a two-stage detector into a single network, which achieves the identification and localization of corn pests in an image. However, in this work, we have further designed coordinate attention-based feature pyramid network by introducing coordinate attention, which can which can obtain the direction-aware and position-sensitive features, leading to detection accuracy improvement. This is different from the previous attention mechanism-based feature pyramid network. Besides, in this work, for network training, we also developed a dynamic positive and negative pest sample selection strategy by adopting a category-specific Gaussian-shaped function. Thus, the proposed method is completely different from the previous work in [1]. As the same time, we have added the citation of the [1] in the introduction (Line 75-80). Thanks again for your suggestion.
Question4: In response to my previous query, "While the manuscript mentions gains in the detection of small-sized pests, it still needs to discuss how the model will behave under a variety of pest sizes and densities—a necessary basis for practical applications. " The authors should consider other datasets.
Answer 4: We are very sorry that we did not understand your previous meaning well, and we have once again answered the question. A large-scale pest detection dataset- IP102, has been used to evaluate the performance of the proposed method. The IP102 dataset contains 102 types of pest, and 75222 images for image classification and 18983 images for object detection task. Thus, in our manuscript, we only select 18983 images for pest detection. We give detailed description of the IP102 dataset in the revised manuscript. Some compared results are carried out, as shown in Table 2.
Table 2 Experimental results on the IP102 dataset
Methods |
AP[0.5,0.95] |
AP0.5 |
AP0.75 |
APs |
APm |
APl |
ARs |
ARm |
ARl |
RetinaNet |
25.9 |
44.8 |
26.7 |
4.2 |
25.7 |
27.1 |
6.8 |
45.6 |
55.4 |
ATSS |
21.2 |
39.1 |
20.3 |
7.8 |
22.0 |
22.2 |
10.5 |
42.4 |
53.6 |
FASF |
23.9 |
44.8 |
22.8 |
8.0 |
27.9 |
25.1 |
6.0 |
44.3 |
51.2 |
FCOS |
11.5 |
22.9 |
10.1 |
2.8 |
15.3 |
12.1 |
5.9 |
38.6 |
49.9 |
FreeAnchor |
26.9 |
46.1 |
27.7 |
3.4 |
27.6 |
28.0 |
4.9 |
46.9 |
56.0 |
Our method |
29.8 |
49.7 |
29.2 |
8.9 |
27.9 |
29.3 |
11.6 |
49.1 |
59.4 |
Finally, we tried our best to improve the manuscript and made some changes in the manuscript. And here we did not list the changes but marked in red in the revised paper. We appreciate for Editors’ and Reviewers’ warm work earnestly and hope that the correction will meet with approval. We look forward to hearing from you regarding our submission. We would be glad to respond to any further questions and comments that you may have.
References
[1] X. Wu, C. Zhan, Y. -K. Lai, M. -M. Cheng and J. Yang, "IP102: A Large-Scale Benchmark Dataset for Insect Pest Recognition," 2019 IEEE/CVF Conference on Computer Vision and Pattern Recognition (CVPR), Long Beach, CA, USA, 2019, pp. 8779-8788, doi: 10.1109/CVPR.2019.00899.

Round 3
Reviewer 3 Report
Comments and Suggestions for Authors
Thanks for your responses. I have a few suggestions for improvement:
Expand on the practical implications of the results, considering a variety of pest sizes, densities, and environmental conditions.
Improve the discussion on novelty by explaining in detail how each new component overcomes certain limitations of previous work.
Provide more qualitative examples and analyses, such as failure cases or conditions where the model struggles.
Could you discuss alternative strategies for mitigating dataset imbalance and compare their potential impact with the current approach?
Also, a more substantial justification of the evaluation metrics and their relevance to real-world scenarios should be included.
Thank you for addressing the previous comments. However, I want to point out that the response includes statements such as, "In order to solve your problem, we conducted relevant comparative experiments on the publicly available large-scale dataset IP102."
It is essential to maintain a neutral and professional tone in the manuscript and responses to reviewers. Phrases like "your problem" personalize the discussion unnecessarily and may be perceived as unprofessional. I suggest rephrasing such comments to ensure objectivity and professionalism.
Author Response
Dear Editors and Reviewers:
Thank you for your letter and reviewers’ comments concerning our manuscript entitled “Precise Agricultural Pest Detection via a Co-ordinate Attention-Based Feature Pyramid Module” (ID: insects-3219756) which is submitted on the journal of Insects. We have carefully studied your question and provided the following answers, hoping to meet your requirements.
Question 1: Expand on the practical implications of the results, considering a variety of pest sizes, densities, and environmental conditions. Improve the discussion on novelty by explaining in detail how each new component overcomes certain limitations of previous work. Provide more qualitative examples and analyses, such as failure cases or conditions where the model struggles.
Answer 2: Thanks for your comments. From the detection results on Agripest 21 dataset and IP102 dataset, we can draw that the proposed method can be used to the detection of pest with various scales, and density. As the same time, the detection speed can achieve 25.8FPS, which can meet practical application requirements.
In this paper, we first developed the coordinate attention-based feature pyramid network to address the feature extraction of pest targets with small size. In addition, some training samples, including positive and negative samples need to be selected. The original IoU-based sample selection method tends to assign negative labels for small pest instances, which cause small-scale pests to be ignored during training. In this paper, by taking the prior center into consideration, we adopt a dynamic sample selection strategy to avoid the usage of IoU-based sample selection methods. The ablation experimental results prove the effectiveness of the proposed method.
From the analysis of the IP102 dataset, we can know that the detection task of IP102 exists the problem of few-shot, which cannot be well addressed by our method. Thus the detection accuracy of our method on IP102 dataset is next to that of AgriPest21. However, for the small pest detection problem that this work mainly aims to solve, our method can effectively solve it. We have also provided experimental records in our current experiments to verify. We also added more failure cases of the model on the IP102 dataset, as shown in Figure 1. From it, we can see that, the detection results of the model have missed detection or false detection. For example, in figure 1(d), the pest “grub” can not be detected because of the posture and shape of the pest. Besides, the pest “Asiatic rice borer” is classified as “yellow rice borer” in Figure 1(e), and the pest “brown plant hopper” is mistaken as “small brown plant hopper” in Figure 1(g). This may because the high similarity among these types of crop pests.
Question 2: Could you discuss alternative strategies for mitigatin dataset imbalance and compare their potential impact with the current approach?
Answer 2: Thanks. The recent methods to address the problem of dataset imbalance can be divided four methods. First, re-sampling method mainly achieves sample balance on the training set, such as oversampling the class samples in tail or undersampling the class samples in head. The resampling based solution is suitable for detection frameworks, but it may lead to increased training time and overfitting risks for tail categories. Second, re-weighting: mainly in training loss, different weights are set for different categories of loss, and larger weights are set for tail category loss. However, this method is highly sensitive to hyperparameter selection and is not suitable for detection frameworks due to its difficulty in handling special background classes (a large number of categories). Third, learning strategy. There are learning methods specifically designed to solve small sample problems that can be referenced, such as meta learning, metric learning, and transfer learning. In addition, the training strategy can be adjusted to divide the training process into two steps: the first step is to train the model normally without distinguishing between head samples and tail samples; The second step is to set a small learning rate and fine tune the model of the first step using various sample balancing strategies. Finally, comprehensive use of the above strategies. For the data imbalance problem, we will focus more on solving the problem of data imbalance in our future work.
Question 3: Also, a more substantial justification of the evaluation metrics and their relevance to real-world scenarios should be included.
Answer 3: Thanks. The mean average precision (mAP) and average recall (AR). To be specific, the metric mAP includes AP0.5, AP0.75, and AP[0.5:0.95], which are computed across various IoU thresholds. Meanwhile, we also examined APs, APm, and APl to assess the detection accuracy of small, medium, and large pests, respectively. Here, the scales are defined as follows: small (area: < 322 pixels), medium (322 pixels ≤ area: < 962 pixels), and large (area: ≥ 962 pixels). Besides, the number of parameters, the GFLOPs, and frame per second (FPS) are used to verify the efficiency of our modules. The above evaluation indicators are currently commonly used indicators for evaluating object detection modules, and they are also sufficient to evaluate our method.
Question 4: Thank you for addressing the previous comments. However, I want to point out that the response includes statements such as, "In order to solve your problem, we conducted relevant comparative experiments on the publicly available large-scale dataset IP102."
Answer 4: We are so sorry for this statement. We will ignore similar statements.
Question 5: It is essential to maintain a neutral and professional tone in the manuscript and responses to reviewers. Phrases like "your problem" personalize the discussion unnecessarily and may be perceived as unprofessional. I suggest rephrasing such comments to ensure objectivity and professionalism.
Answer 5: We apologize again for our inappropriate expression and will pay more attention to our statements in the future. Thanks.
